# Explain My Surprise: Learning Efficient Long-Term Memory by Predicting Uncertain Outcomes

**Artyom Sorokin**
AIRI, MIPT
Moscow, Russia
asorokin@airi.net

**Nazar Buzun**
AIRI
Moscow, Russia
buzun@airi.net

**Leonid Pugachev**
MIPT
Dolgoprudny, Russia
puleon@mail.ru

**Mikhail Burtsev**
AIRI, MIPT
Moscow, Russia
burtsev@airi.net

## Abstract

In many sequential tasks, a model needs to remember relevant events from the distant past to make correct predictions. Unfortunately, a straightforward application of gradient based training requires intermediate computations to be stored for every element of a sequence. This requires to store prohibitively large intermediate data if a sequence consists of thousands or even millions elements, and as a result, makes learning of very long-term dependencies infeasible. However, the majority of sequence elements can usually be predicted by taking into account only temporally local information. On the other hand, predictions affected by long-term dependencies are sparse and characterized by high uncertainty given only local information. We propose MemUP, a new training method that allows to learn long-term dependencies without backpropagating gradients through the whole sequence at a time. This method can potentially be applied to any recurrent architecture. LSTM network trained with MemUP performs better or comparable to baselines while requiring to store less intermediate data.

## 1 Introduction

Two dominating approaches for memory augmentation in deep learning include recurrent neural networks [1; 2; 3; 4; 5; 6; 7; 8; 9]) and Transformers [10; 11; 12; 13; 14; 15]. Historically, recurrent networks and Transformers have been primarily developed for applications in natural language processing, which is characterized by strong dependencies between closely located elements. On the other hand, in the reinforcement learning setting, the agent often has to remember only a few bits of information but for a very long time [16].

Evolution of neural architectures tackling the problem of learning long-term dependencies follows the path of increasing size and complexity [6; 8; 17]. For example, Transformers[18] and memory augmented neural networks [7] perform much better than the classical LSTM in tasks with long-term temporal dependencies. Still, even gigantic language models like GPT-3 [19] process sequences only up to 2048 elements due to quadratic complexity of self-attention. So, these architectures are much more demanding in terms of the memory capacity and computational power required for training.

To learn a long-term dependency between a distant observation in the past and agent's actions in the current state, both the recurrent networks trained with back propagation through time and Transformers need to store gradients for all intermediate steps. Taking into account, that the minimum distance between useful information in the past and the moment of its utilization can be measured in thousands or millions of time steps, such solutions seem unrealistic given the current hardware capabilities. We propose a training method that allows an agent to find and store dependencies between temporarily distant events without the need to process intermediate steps.

Our contributions are the following:

36th Conference on Neural Information Processing Systems (NeurIPS 2022).

- We propose a new formulation for the problem of memory learning as maximisation of conditional mutual information between memory state and future outcomes.
- We introduce an original method MemUP (**Mem**ory for **U**ncertainty **P**rediction) for efficient long-term memory training by minimizing uncertainty of predictions for states where local information is not enough.
- We show that implementation of our method for recurrent neural networks is able to learn long-term dependencies better than baselines while propagating gradients only through a small fractions of a sequence in one optimization step. Notably, the applications of MemUP are successful in supervised learning tasks with different loss functions as well as in reinforcement learning tasks.

## 2 Memory to explain surprise

The main idea of our approach is very simple. Find the most surprising events or elements of a sequence and train a memory to detect and store information that allows to explain out these events. Below we formulate this idea in the framework of information theory.

Lets consider a sequence $\tau$ with inputs $\{x_t\}_{t=0}^T$ and targets $\{y_t\}_{t=0}^T$. A model is trained to predict targets given inputs. An ideal memory could just store all previous inputs $\{x_i\}_{i=0}^t$ at each step $t$ and predict $y_t$ from them, but it is too computationally expensive. In practice a memory model $g_\theta$ has to selectively store some aggregation of this sequence in a memory state $m_t = g_\theta(x_t, m_{t-1})$ . Then it should be decided what to write and what to remove from the memory.

**Estimating past information importance.** Consider an arbitrary time step $k$. The past element $x_t$ has useful information about the target $y_k$, if $y_k$ depends on $x_t$ given current $x_k$, i.e. $p(y_k|x_k, x_t) \neq p(y_k|x_k)$. In this case, we can say that there is a temporal dependency with the length of $k - t$ steps, that starts at step $t$ and ends at step $k$. The strength of temporal dependency or importance of remembering $x_t$ can be measured as an amount of change in the distribution of $y_k$ given a knowledge of $x_t$ . Multiplication of the both parts of the inequality above by $p(x_t|x_k)$ and application of KL-divergence as a measure of the difference between distributions leads to the conditional mutual information as a measure of $x_t$ importance:

$$I(y_k; x_t|x_k) = \mathbb{E}_{x_k}\mathcal{KL}[p(y_k, x_t|x_k)\|p(y_k|x_k)p(x_t|x_k)], \tag{1}$$

where $x_k$ is taken from sequences stored in some dataset or generated by an agent over interaction with the environment.

**Objective function for memory training.** Then, the problem of learning what to store in the memory can be defined as maximization of the mutual information between memory states and prediction targets with respect to the parameters $\theta$:

$$\max_\theta \sum_{k=t}^T I(y_k; m_t = g_\theta(x_t, m_{t-1})|x_k). \tag{2}$$

The sum in eq. 2 allows us to process each memory update in a separate gradient step. As accounting for all future steps prevents loss of information that can be helpful in the distant future. Unfortunately, this still requires the entire remaining (future) sequence to be processed for every update and has the complexity $O(T)$ with respect to the sequence length.

**Reducing the cost of memory training.** Fortunately, updating the memory for all future steps is not necessary for the majority of real-world tasks. As the distance between events increases, the number of dependencies also decreases. This phenomenon is sometimes called locality of reference [20]. In other words, information from the greater part of past inputs may be useless for $y_k$ prediction given $x_k$. Accordingly, most of the elements in the sum (eq. 2) will be close to zero. Therefore, depending on the structure of the problem, we could vary the number of elements in the sum, leaving only those for which long-term memory is critical for correct predictions. The main problem is to find such steps in the sequence.

Lets assume that the ideal memory $m_t^*$ that stores all possible information from the past is available. Then the conditional mutual information at a step $t$:

$$I(y_t, m_t^*|x_t) = H(y_t|x_t) - H(y_t|m_t^*, x_t), \tag{3}$$

shows the potential utility of memory for prediction at this step. When the number of elements in the sum in eq. 2 is minimized to reduce computations, the steps with the highest value of mutual information from eq. 3 should be preserved to assure good quality of prediction. Naturally, we cannot estimate $I(y_t, m_t^*|x_t)$ directly, because $m_t^*$ is not available to us.

As seen from the eq.3, $m_t^*$ participates only in the second entropy, which indicates how well one can predict the future in the presence of ideal memory and ideal model. On the other hand, the local entropy $H(y_t|x_t)$ can be easily estimated. A value of local entropy would be sufficient for our task if we could use it to determine the ordering for values of the potential memory utility $I(y_t, m_t^*|x_t)$. Specifically, for every pair of elements $i$, $j$ if $H(y_i|x_i) > H(y_j|x_j)$ then $I(y_i, m_i^*|x_i) > I(y_j, m_j^*|x_j)$. This becomes true under one of two possible conditions: (1) $H(y_t|x_t) \gg H(y_t|m_t^*, x_t)$ or (2) $H(y_t|m_t^*, x_t) \sim \epsilon$. The first condition means that given a perfect memory and a model, one could make high-quality predictions for the task at hand. The second condition means that the real distribution of the target variable $y_t$ has a similar amount of noise at each step.

If our task satisfies either one of these two conditions, then valuable information from the past could make the biggest contribution for the elements with the highest values of average "surprise" $H(y_t|x_t)$.

We should note that estimation of $I(y_j, m_j^*|x_j)$ by the local entropy makes MemUP vulnerable to the Noisy-TV problem[21], when the conditions we have described are not satisfied. Sensitivity tests for the Noisy-TV problem are discussed in Section 7.

Thus, instead of optimizing the memory for each future step, we introduce a new objective function that allows us to train it only for steps where the potential gain from the past information is maximal:

$$\max_\theta \sum_{k \in \mathcal{U}_t} I(y_k; m_t = g_\theta(x_t, m_{t-1})|x_k), \qquad (4)$$

here $\mathcal{U}_t$ denotes the set of top-$K$ indices of steps from $t$ to $T$ with the highest estimated local entropy $H(y_k|x_k)$. The hyperparameter $K$ controls fraction of a sequence to be processed in one gradient step. On the other hand, maximizing the mutual information between the memory $m_t$ and an arbitrary distant event at step $k > t$ allows to learn long-term dependencies.

**Optimization.** Directly optimizing mutual information can be a challenging task. In our case, we use variational lower bound on mutual information proposed by [22]. Using the lower bound we can maximize mutual information by minimizing Cross-Entropy (CE) between the empirical and model distribution for all selected high entropy events:

$$\min_{\theta, \phi} \sum_{k \in \mathcal{U}_t} \mathbb{E}_{x_k, y_k} \left[ -\log q_\phi(y_k|m_t^\theta, x_k) \right], \qquad (5)$$

where $m_t^\theta$ is a shortcut for $m_t = g_\theta(x_t, m_{t-1})$, $q_\phi$ is a predictor network with parameters $\phi$, that estimates probability of $y_k$ given $x_k$ and $m_t$. For a detailed derivation of eq. 5 see Appendix A in Supplementary Materials.

Collection of elements with high uncertainty $\mathcal{U}_t$ requires a special uncertainty detector $d_\psi$ model. The main requirement for $d_\psi$ is to produce uncertainty estimates $s_t = d_\psi(x_t)$. In the simplest case, the detector estimates "surprise" $-\log p(y_t|x_t)$ (prediction error), which can be seen as a single point estimate of uncertainty. Other models that directly estimate the uncertainty or a whole distribution [23; 24] can also be used.

## 3 MemUP for Recurrent Neural Architectures

Recurrent neural net implementation of MemUP consists of (1) training of uncertainty detector model $d_\psi$, (2) selection of elements with highest uncertainty $\mathcal{U}_t$ and (3) training memory $g_\theta$ and predictor $q_\phi$ models. A pseudocode for RNN MemUP training is shown in Appendix G.

**Uncertainty detector training.** Information maximization reasoning does not limit the choice of solutions for uncertainty detector $d_\psi$. In our experiments on algorithmic tasks we use a recurrent neural classifier trained with Cross-Entropy Loss. In this case detector's "surprise" $-\log d_\psi(y_t|x_t)$ is used as an uncertainty estimate.

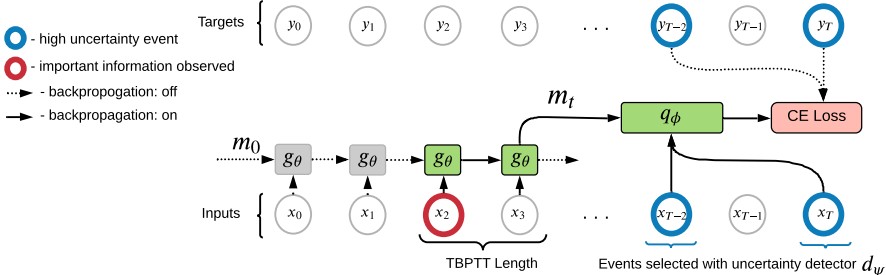

Figure 1: MemUP gradient update. The recurrent memory $g_\theta$ processes a sequence. The number of prediction targets $K = 2$ and TBPTT length $r = 2$. Blue circles denote states with the highest uncertainty estimates $d_\psi(y_k|x_k)$. Here $x_2$ marked by red circle contains information that can help to predict outcomes $y_{T-2}$ and $y_T$. At the end of RNN TBPTT rollout these states are selected to form a set $U_3 = \{T - 2, T\}$. Then, $m_3$, $x_{T-2}, y_{T-2}, x_T, y_T$ are used to compute CE-Loss according to eq. 5. While states 2 and T could be separated by many millions steps, MemUP allows memory to capture the utility of information from $x_2$ by propagating gradients only through the sequence of $K + r = 4$ elements at a time.

In Reinforcement Learning experiments we use distributional algorithm QRDQN [24] trained to predict discounted future returns $R_t$ ($y_t = R_t$ in RL experiments). As QRDQN approximates a whole distribution of the future returns, estimating uncertainty becomes a simple task.

**Processing a sequence with the memory network.** MemUP allows to learn long term-dependencies without propagating gradients through the whole sequence. To demonstrate that in our experiments we use Truncated BPTT to train memory network $g_\theta$. In the majority of supervised learning experiments we set rollout length $r$ between 10 and 60 steps. In all Reinforcement Learning experiments $r = 1$, i.e. recurrent memory is trained without actually using backpropagation through time. In All Experiments $g_\theta$ is a simple LSTM network with additional input encoder.

**Selecting states with high uncertainty.** Given uncertainty estimates $\{s_i\}_{i=t}^T$ for each future element in the sequence, we use softmax sampling $p(k) = e^{s_k/\tau} / \sum_{i=t}^T (e^{s_i/\tau})$ without replacement with $\tau = 0.02$. To implement sampling without replacement Gumbel-Max Trick [25] is used. A scheme illustrating a single gradient update is shown in Figure 1.

## 4    Related work

The problem of learning long-term dependencies has long been studied in both supervised learning [26; 27] and reinforcement learning domains [28]. However, most of the research related to recurrent networks architectures is aimed at solving the problems of exploding and vanishing gradients [2; 29; 7; 30]. The common trend is that many of the existing solutions to deal with vanishing gradients simultaneously increase the cost associated with the calculation of these gradients [29; 7]. For example, Transformers[10] generally require to store $O(N^2)$ of intermediate results for training on sequences of length $N$. The recently proposed Linformer [31] and BigBird [20] architectures allow training Transformers with linear complexity in space. However, most of these new Transformers have not yet found their way in the reinforcement learning setting.

In the field of deep reinforcement learning, the dominant approaches to implementation of memory are based on recurrent neural networks in combination with modern reinforcement learning algorithms [5; 32; 33; 34]. There are many studies that propose memory architectures for specific features of the reinforcement learning problems [9; 8]. In the paper by Parisotto and Salakhutdinov [35], a specifically modified DNC [7] architecture learned long-term dependencies of several hundred steps in the setting of 2D and 3D navigation tasks. The downside of this architecture is that the agent needs access to the information about its absolute or relative location in a 2D/3D environment. In another work, Ha and Schmidhuber [36] used a complex procedure with pre-training of memory and state embeddings. Their overall learning procedure is similar to ours. However, their memory/world model is trained to make predictions about the next step observations only, while we train memory to make predictions about events that can be arbitrarily far in the future.

Wayne at al. [17] demonstrated the state of the art results with the MERLIN algorithm. MERLIN's memory kept observation embeddings trained with a complex variational autoencoder in a DNC-

like [7] list. The memory module was also trained separately from the policy. Hung et al. [37] followed up on the MERLIN architecture's success and used the soft attention mechanism over all past embeddings to encourage exploratory actions in the POMDP environment. Their experiments showed models learning temporal dependencies with a length of 500+ steps. In contrast to our work, both of these solutions still require processing an arbitrarily long sequence of state embeddings by the policy networks at each step.

The direct attempts in applying Transformer architecture in reinforcement learning setting resulted in conclusion that Transformers are too unstable to work properly in RL [38]. Parisotto et al. [18] were able to overcome the instability of Transformers, showing state of the art result in the memory dependent 3D tasks. Another recent work, by Loynd et al. [39], has also applied Transformers to the POMDP problems, but the WMG architecture relies on availability of factored observations to the agent which imposes additional requirements on the environment. Switching focus from Transformers and complex architectures, Beck et al. [16] proposed AMRL architecture: a simple modification to the LSTM architecture to combat the problem of learning long-term dependencies in environments with a significant amount of observational noise.

## 5   Supervised Learning Experiments

For evaluation and comparison of our method we use four tasks: Copy [40], Scattered copy, Add [2] and permuted sequential MNIST (pMNIST) [41]. All these tasks are benchmarks that are used for testing models with long-term memory. In the original Copy task a sequence of length $l$ ($l = 10$ in out experiments) should be copied after a $T - l$ steps ($T \in \{120, 520, 1020, 5020\}$ in our experiments), and in Scattered copy task a model has to make predictions in locations inside range $[l, T-1]$ that are chosen at random. We add this task in order to make detection of high-uncertainty locations harder. For Copy, Scattered copy and Add tasks train and test datasets have sizes 10K and 1K, for pMNIST 60K and 10K. For detailed tasks description see Appendix B.

In these experiments predictor $q_\phi$ consists of a three-layered MLP and a recurrent input encoder. The encoder $E^{rnn}$ consists of a single fully-connected layer followed by two LSTM-layers with 128 hidden units and dropout probability $0.1$. $E^{rnn}$ is used to encode bigger local context than original inputs corresponding to indices in the $U_t$ set (it is also possible to use a feedforward network for this task). Memory $g_\theta$ has the same architecture as $E^{rnn}$ with separate weights.

We do not train a separate detector $d_\psi$. Uncertainty at each step is estimated by prediction error from the predictor module $q_\phi$. For more information you can look at our implementation ( link in Appendix E).

In the Add Task the model is trained with MSE loss, which is a special case of CE loss under the model-assumption that target distribution is a Gaussian with unit variance. Both memory and encoder are trained with the same Truncated BPTT length. Truncation length equals 10 in all tasks with sequence length $< 1000$, 20 in tasks with length $\geq 1000$ and 30, 60 in pMNIST 784, 3136 respectively.

We compare MemUP with LSTM [2] and Transformer [10] models from pytorch library, as well as a recurrent network SRNN [30] which is designed for training on long sequences and has reduced saturation of gradients. LSTM* and SRNN* have the same architecture as $E^{rnn}$ with 3 layered

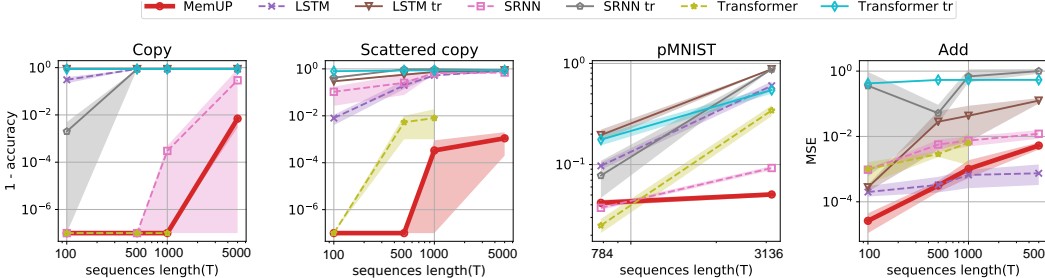

Figure 2: Final performance on supervised learning tasks. X-Axis shows task scaling, while Y-axis correspond to metric score at the end of training. Metrics: Inverted Accuracy (1. - Accuracy) in Copy, Scattered copy and pMNIST tasks, MSE in Add task. All curves are averaged over 3 runs.

MLP on top. In the case of SRNN baselines, we have replaced all LSTM layers with SRNN layers. The Transformer model consists of a single FC layer followed by 4 self-attention layers and the same 3 layered MLP on top. Baselines were trained in two modes. In the first mode (columns with Transformer, SRNN, LSTM) all elements of input sequence of length $T$ were fed into a model simultaneously, so gradients propagate through the whole sequence during the training. We don't have results for Transformer on sequences of 5000+ steps since it didn't fit into our GPU RAM. In the second mode (Transformer Tr., SRNN Tr., LSTM Tr.) BPTT rollout for recurrent architectures and attention window for Transformer were truncated to process the same number of steps as MemUP ($r + K$).

Results in Figure 2 show that only MemUP allows to achieve a good quality of prediction in all tasks. In truncated BPTT setting (solid lines) MemUP is significantly better than LSTM tr, SRNN tr or Transformer tr due to MemUP ability to learn temporal dependencies longer that the length of BPTT rollout. When full information about the sequence is available to LSTM, SRNN and Transformer (dashed lines) but not MemUP, the latter still shows the best results for longest versions of Copy, Scattered Copy, pMnist, and has better overall performance.Moreover, when compared with Full BPTT baselines, MemUP requires to store much less intermediate computations (10-250 times) due to usage of short Truncated BPTT rollouts.

## 6  Reinforcement Learning Experiments

Performance of the MemUP algorithm is studied in two reinforcement learning tasks. The first task is classic T-Maze environment where the main difficulty is a long-term dependency between a hint at the starting position and location of a reward at the exit (see Figure 3.1). The main advantage of this problem is that it allows to test the agent's memory mechanisms in isolation because of primitive policy and the observation space. We base our experiments on the noisy version introduced by [16].

The second task is the color dependent object collection in the Doom environment [42; 43] which requires long-term memory in combination with a more complex policy and rich observation space. In the task introduced by [44], the agent is placed in a room filled with acid (see Figure 3.2.) and constantly loses health. Objects of green and red colors are scattered throughout the environment. Object of one color replenish the agent's health and give a +1 reward, while others take away health and give a -1 reward. The correspondence between effects and colors is determined randomly at the beginning of each episode. This information is conveyed by a column whose color matches the color of health replenishing items. In our version the column disappear after first 45 steps (see Figure 3.3). The episode ends when the agent's health drops to zero or after 1050 time steps.

The most straightforward extension of the MemUP algorithm to a reinforcement learning problem is to use memory state $m_t$ as an additional input to an RL agent. There are several possible solutions to combine MemUp's training with agent's policy training: (1) pre train MemUP and then train an agent, (2) use alternating memory and agent training phases, train memory and agent in parallel, (3) combining the MemUP and RL agent into a single into an end-to-end network. In these experiments, we test the simplest version: train MemUP on trajectories generated with a frozen policy, then train an agent with a frozen MemUP memory. The second version is also implemented in the code (link in Appendix E).

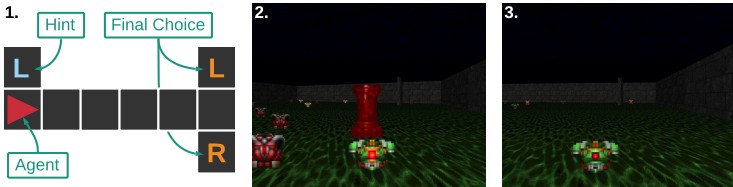

Figure 3: Environments for testing long-term time dependencies. **1.** *T-Maze environment.* The agent should reach the T-shaped junction and choose one of the arms (L or R). A hint about what arm to choose is provided at the very beginning. **2.** and **3.** *Vizdoom-Two-Colors environment.* The agent is in the room and constantly loses health. To replenish his health and recieve a reward the agent needs to collect items of the same color as the column. The color of the column is chosen randomly from red or green options at the beginning of the episode. After 45 steps from the beginning the column disappears. The episode lasts for a maximum of 1050 steps.

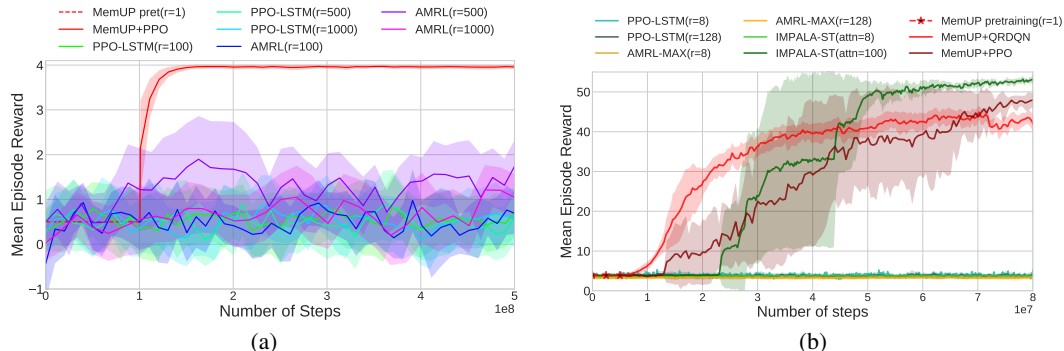

Figure 4: **a)** Learning curves for MemUP and baselines in T-Maze-LNR-1000. **b)** Learning plots in the Vizdoom-Two-Colors environment for all agents. All curves are averaged over 3 runs. Translucent areas around each curve show the standard deviation computed over all runs. The memory pre-training phase of MemUP agent is marked with the dashed line. TBPTT length/attention span for each baseline is specified in the parenthesis.

Inputs $x_t$ are formed from observations $o_t$ and previous action $a_{t-1}$. This time Observations are encoded using convolutional networks or fully connected networks. We use discounted future returns $R_t = \sum_{i=t}^{T} \gamma^{i-t} r_i$. as targets $y_t$. In the experiments, MemUP is compared with the following baselines:

**PPO-LSTM** is a recurrent version of PPO [45] with a single LSTM-layer [2]. We use PPO-LSTM implementation from the RLPyt library [46].

**IMPALA-ST** is an IMPALA [47] agent using Stabilized Transformer[10] architecture. Stabilized Transformer was presented by Parisotto et al. [18]. For experiments, we use the only available open implementation of this algorithm [48].

**AMRL** is proposed by [16] for the Noisy T-Maze Task. In the original study AMRL was compared with many different memory architectures including DNC, multi-layered LSTM and outperformed all of them on T-Maze-LN-100 task as well as other memory experiments. AMRL is similar to PPO-LSTM baseline, but extends LSTM with AMRL Layer.

In all experiments we use the same encoder networks for observation embeddings for all baselines and MemUP. Additionally the size of an LSTM-layer is the same for the memory module $g_\theta$, PPO-LSTM and PPO-AMRL baselines. Results for the T-Maze-LNR-1000 (10 times longer dependency than in [16]) environment are presented in Figure 4a. The designation T-Maze-LNR-1000 means that the maze length (and corresponding tempral dependency) in each episode can be from 1000 to 1009 steps. Several rollout lengths are tested for RNN-based baselines. We could not successfully train a single PPO-LSTM or AMRL agent even with TBPTT rollout of $r = 1000$[1].

During MemUP pre-training phase, episodes are generated with a random policy. We use the discounted future reward with $\gamma = 0$ as a prediction target. A policy training for the MemUP agent starts after the memory pre-training phase is completed. The time spent in the pre-training phase is marked with a dashed line in all figures. As shown in Figure4a the agent learns almost instantly, given a memory pre-trained with our method. For the MemUP agent we train all components including memory module and uncertainty detector for each individual run from scratch. Additionally, Appendix Dshows that MemUp can be trained to solve T-maze-LNR-20000.

Results for the Vizdoom-Two-Colors environment are shown in Figure 4b. IMPALA-ST(attn=8) with a short attention span could not learn any reasonable policy. On the other hand, IMPALA-ST(attn=100) solves the problem and reliably survive in the environment for 1050 steps. Both PPO-LSTM and AMRL baselines with TBPTT rollouts of 128 and 8 steps could not learn to survive longer than a random agent. While the rollout of 128 steps is potentially enough to remember the color of the column or a previously collected items, PPO-LSTM and AMRL agents are unable to utilize long-term information from the past.

---

[1]We tested our AMRL implementation on T-Maze-LNR-100 with full rollout and get results aligning with the original paper. But the algorithm did not withstand 10 times increase in episode length and truncation constraints

For MemUP we use the discounted future reward with $\gamma = 0.8$ as a prediction target. The memory module $g_\theta$ is trained with the rollout length of 1 step. In Vizdoom-Two-Colors we set number of targets $K = 3$. Thus, to train the memory $g_\theta$ we use 4 separate observations per episode for one gradient update. In ViZDoom, we train PPO (MemUP+PPO) and QRDQN (MemUP+QRDQN) agents with MemUP pretraining.

Both MemUP+PPO and MemUP+QRDQN learn to survive in the environement for full 1050 steps by collecting healing items. Though the final performance of IMPALA-ST(attn = 100) is slightly better. We speculate that this is due to the fact that the pre-trained memory **M** has learned to store mostly information about the color of the column, while IMPALA-ST (attn=100) has slightly better spatial awareness. On the other hand, MemUP use significantly fewer resources in terms of the size of processed sequences during training/pre-training and evaluation. As well, MemUP significantly outperforms baselines with comparable resource requirements, like IMPALA-ST (attn=8), PPO-LSTM(r=8) and AMRL-MAX(r=8).

## 7 Sensitivity to the Noisy-TV problem

In section 2 we discussed conditions on true uncertainty of targets $y_t$ given a perfect memory $m_t^*$. The goal was to set conditions under which we can make a good guess about relative values of potential memory utility $I(y_t|m_t^*, c_t)$ by estimating local entropy $H(y_t|c_t)$. However, this may not be the case. For example, some targets $y_t$ may be completely unpredictable and no information from the past can reduce the entropy for the distribution of $y_t$: $H(y_t|x_t) \sim H(y_t|m_t^*, x_t)$. Then the local entropy is large, while potential utility of memory for $y_t$ predictions is close to zero. Training memory to predict such targets is pointless and prevents MemUP from focusing on events for which training long-term memory is essential. The problem of over-emphasis on unpredictable surprising events is called Noisy-TV problem[21]. Noisy-TV problem is often encountered by algorithms using curiosity-based exploration in reinforcement learning[49]. To test robustness of MemUp to the Noisy-TV problem, we have modified T-maze-LNR-100 environment.

**Noisy T-maze With Distractors.** In this version, agent can receive +4 or -3 rewards in D+1 decision points. Their location in the corridor is chosen randomly at a distance of at least 50 steps from the hint. All of them can be detected when $o_t[1]! = 0$. element in observation vector $o_t$. Decision points are distinguishable from each other. Each decision point has a unique value of $o_t[1]$. Only in one decision point, the next reward depends on the agent's action and the value of the hint ($o_t[1] = 1$. in this case ). In other D decision points, the next rewards are completely random ( chosen with a probability of 0.5 ). Thus, Noisy T-maze With Distractors generate sequences with one long-term dependency and D events acting as Noisy-TV distractors.

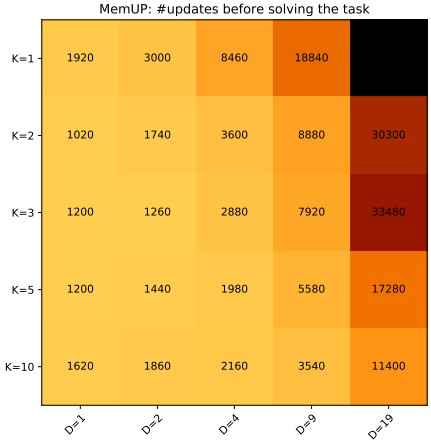

Figure 5: Each cell shows the mean number of updates required for the memory to correctly predict (achieving $0.1$ RMSE) the reward after the decision point connected with the temporal dependency. The x-axis shows the number of unpredictable decision points in the environment. The Y-axis shows number of events predicted by MemUP. Darker colors means slower solution.

We conducted 25 experiments with 6 independent runs in each (150 runs in total). The differences in the experiments come down to two parameters: number of distracting events D (X-axis in Fig. 5) and number of events sampled for prediction K (Y-axis in Fig. 5). More distracting events make the task of learning the temporal dependency harder. Increasing the size of $\mathcal{U}_t$ set( by changing $K$) improves the chance of selecting a future event that is a part of temporal dependency. On the other hand, with an increase in $\mathcal{U}_t$, the resource cost for memory training also increase. The TBPTT rollout length in these experiments is 1.

Results in Fig. 5 show that with an increase in the number of distractions, memory learning speed decreases. However, increasing the set $\mathcal{U}_t$ allows to alleviate this problem. For the experiment (D=19, K=1), none of the runs solved the task in the first 45000 updates. In experiments (D=19, K=2) and (D=19, K=3) only 4 out of 5 runs solved the task. The results show that the agent can learn the time dependency fairly quickly even if the uncertainty detector select noisy events $80\%$ of the time.

## 8 Ablation study

To study an effectiveness of our core idea, i.e. training memory by predicting long-term future events with high uncertainty, we compared MemUP with baselines that exclude core MemUP features while sharing the same neural architecture.

In these experiments, we carefully consider the process of learning memory (without further training the RL agent) on the T-Maze-LNR problems. The following memory ablation baselines were studied:

**MemUP:** Proposed in this paper. See section 2.

**Rnd-Pred:** Same as MemUP, but steps for future outcome prediction are selected randomly and uniformly among all future timesteps. This randomization verify whether predicting steps with high uncertainty helps MemUP to overcome the problem of learning long-term dependencies with small BPTT rollouts.

**Default:** Same as MemUP, but it is trained to predict return $R_t$ at each step $t$, as oposed to an arbitrarily distant future events as in MemUP and *Rnd-Pred*. Thus, *Default* is the same as the LSTM baseline in the Supervised Learing Experiments (see 5). But in this study we test this baseline on all intermediate Truncated BPTT lengths.

All versions have the same architecture, and hyperparametes. They were trained in T-Maze-LNR-100 and T-Maze-LNR-1000 problems. Episodes were generated by a random strategy. All versions are trained with 5 different Truncated BPTT lengths: 1, 5, 10, 50, 100. Testing metric is a root-mean-squared error(RMSE) between the models' prediction and the actual return at the final step of the episode. The evaluation was conducted on the 100 separately recorded episodes.

Figure 6 shows the results of the experiments. On the x-axis, we have specified the length of the Truncated BPTT in decreasing order, i.e. in the order of increasing task complexity. For every rollout length there is a corresponding absolute value of error between predictions of the model and targets

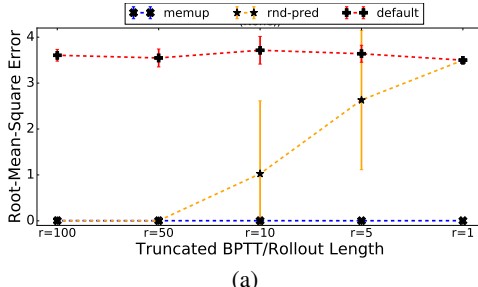
(a)

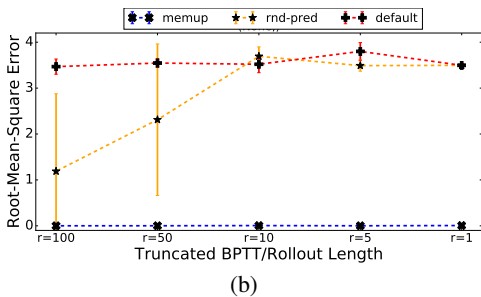
(b)

Figure 6: Averaged over 6 runs prediction error at the end of training for each ablation baseline with respective TBPTT length. Vertical bars show standard deviation computed over 6 runs. **a)** T-Maze-LNR-100. **b)** T-Maze-LNR-1000.

at the end of training ($3e4$ updates for T-Maze-LNR-100 and $3e5$ for T-Maze-LNR-1000). Smaller values represent better predictions.

The *Default* modification is unable to learn a long-time dependency exceeding the length of the tested rollouts. This is the same as the results of the PPO-LSTM agent on the T-Maze-LNR Reinforcement Learining experiments.

The *Rnd-Pred* modification solves the task with rollouts 100 and 50 in T-Maze of length 100, but for shorter rollouts and for longer T-Maze-LNR-1000 results deteriorate significantly. In the majority of runs in T-Maze-LNR-1000, the memory has not been learned. The relative success of *Rnd-Pred* is expected because random selection of prediction target still allows to sample a useful future event with some small probability.[2] Thus, random sampling of prediction targets allows to learn casual dependency that is 2 times longer than TBPTT, but already struggles at ratio of 10.

On the other hand, the MemUP is able to solve the problem regardless of the rollout length, i.e. MemUP demonstrated an ability to learn casual dependencies that 1000 times longer than its TBPTT length in this setting. In this study MemUP is successfully trained with TBPTT of length 1, in other words, without using backpropagation through time at all.

## 9 Conclusion

In this paper, we proposed a new method for training long-term memory. The main idea is to train a memory network to predict future outcomes of high uncertainty and skip all others. Predicting a small number of arbitrarily distant future outcomes substantially saves computational resources required to backpropagate gradients. At the same time, the emphasis on high uncertainty outcomes allows not to miss long-term dependencies in the task.

Experimental results show that our training algorithm allows to learn temporal dependencies significantly longer than the number of steps processed for a single gradient update. None of the baseline architectures trained in a classical way demonstrate such ability. All these baselines with comparable results require to store at least 200 times more intermediate calculations for the Add, Copy, Scattered Copy, pMNIST tasks as well as at least 500 times more for the T-maze, and 625 times more for the Vizdoom-Two-Colors (Stabilized Transformer baseline has quadratic complexity while processing 25 times more steps).

Even using fewer resources, MemUP outperforms non-truncated baselines that has simultaneous access to all elements of a sequence (including Transformer) on pMNIST(3136), Copy and Scattered Copy tasks. On T-maze and Vizdoom-Two-Colors tasks, MemUP is better than all recurrent baselines. Another advantage is that MemUP can be combined with any recurrent architecture and applied both to supervised as well reinforcement learning settings.

We believe that the MemUP algorithm applied to recurrent networks has shown promising results, and the main idea demonstrates exciting avenues for future research.

## 10 Acknowledgments and Disclosure of Funding

This work was partially supported by a grant for research centers in the field of artificial intelligence, provided by the Analytical Center for the Government of the Russian Federation in accordance with the subsidy agreement (agreement identifier 000000D730321P5Q0002) and the agreement with the Moscow Institute of Physics and Technology dated November 1, 2021 No. 70-2021-00138. This work was partially supported by the Ministry of Science and Higher Education of the Russian Federation (Grant No. 075-15-2020-801).

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
