# Explain My Surprise: Learning Efficient Long-Term Memory by Predicting Uncertain Outcomes Supplementary Materials

**Artyom Sorokin**
AIRI, MIPT
Moscow, Russia
asorokin@airi.net

**Nazar Buzun**
AIRI
Moscow, Russia
buzun@airi.net

**Leonid Pugachev**
MIPT
Dolgoprudny, Russia
puleon@mail.ru

**Mikhail Burtsev**
AIRI, MIPT
Moscow, Russia
burtsev@airi.net

## A  Mutual Information Maximization

Estimating mutual information $I(y_k, m_t^\theta | x_k)$ can be difficult task from a computational point of view. Therefore, to make the task more trackable we use a lower bound for mutual information derived by [1]. Adapting it for the `MemUP` case, we get the following inequality:

$$
\begin{aligned}
I(y_k, m_t^\theta | x_k) &= H(y_k | x_k) - H(y_k | m_t^\theta, x_k) \\
&\geq H(y_k | x_k) - CE(p(y_k | m_t^\theta, x_k), q_\phi(y_k | m_t^\theta, x_k)),
\end{aligned}
\tag{1}
$$

where $m_t^\theta$ is a shortcut notation denoting dependence of the memory state $m_t$ on the parameters of memory network $g_\theta$. The distribution $p$ is the true conditional distribution of future outcomes $y_k$, which is unknown to us, but we can sample from it using training data. The distribution $q_\phi$ is approximated by the predictor network with parameters $\phi$. Inequality 1 follows from the relation between KL-divergence, cross-entropy (CE) and entropy: $D_{KL}(p||q_\phi) = CE(p, q_\phi) - H(p)$. Using non-negativity property of KL-divergence leads to $CE(p, q) \geq H(p)$. The bound becomes exact when $q_\phi(y_k | m_t^\theta, x_k)$ is equal to $p(y_k | m_t^\theta, x_k)$

Since $H(y_k | x_k)$ is independent from memory and predictor networks, maximizing the lower bound is the same as minimizing cross-entropy. Therefore, the memory $g_\theta$ and the predictor $q_\phi$ can be jointly trained to maximize mutual information by simply minimizing cross entropy loss:

$$
\min_{\theta, \phi} \sum_{k \in \mathcal{U}_t}^T CE(p(y_k | m_t^\theta, x_k), q_\phi(y_k | m_t^\theta, x_k)) = \min_{\theta, \phi} \sum_{k \in \mathcal{U}_t}^T \mathbb{E}_{x_k, y_k}[-\log q_\phi(y_k | m_t^\theta, x_k)].
\tag{2}
$$

Specifically, the memory module $g_\theta$ is trained by backpropagating gradients directly through the predictor network $q_\phi$.

## B  Supervised Learning Tasks

For accuracy evaluation we involve four tasks: Copy, Scattered copy, Add and permuted MNIST (pMNIST). All these tasks are benchmarks that are used for testing models with long-term memory.

**Copy task.**  This is an aligned sequence to sequence classification problem $(X \to Y)$. Denote by $T$ length of $X$ and $Y$. The first 10 elements contain random uniform digits from set $\{2, \ldots, 9\}$ that model has to copy and reproduce in the end of the sequence where $X_i = 1, i \in \{T - 10, \ldots, T - 1\}$. In all other positions except the first 10 and the last 10 $X_i = 1, i \in \{10, \ldots, T - 11\}$. Correspondingly $Y_i = 0, i \in \{0, \ldots, T - 11\}$ except the last 10 where $Y_{T-10+i} = X_i, i \in \{0, \ldots, 9\}$. We split this task into four different sub-tasks by the length where $T \in \{120, 520, 1020, 5020\}$. Train dataset contains 10k sequences and test consists of 1k sequences.

36th Conference on Neural Information Processing Systems (NeurIPS 2022).

Table 1: Supervised Learning Results. **Metrics:** `Accuracy` (classification accuracy in %) in Copy, Scattered copy and pMNIST tasks, `MSE` in Add task. **Bold** font highlights the best score per task, * shows the best score among truncated methods.

| Task/Method | MemUP | LSTM | LSTM tr | SRNN | SRNN tr | Transf. | Transf. tr |
|---|---|---|---|---|---|---|---|
| Copy 120 | **100**\* | 69.3 | 12.7 | **100** | 99.7 | **100** | 12.5 |
| Copy 520 | **100**\* | 12.4 | 13.0 | **100** | 13.1 | **100** | 12.7 |
| Copy 1020 | **100**\* | 12.4 | 12.9 | 99.9 | 12.5 | **100** | 12.6 |
| Copy 5020 | **99.3**\* | 12.4 | 12.5 | 70.8 | 12.5 | out of mem | 12.5 |
| Scat. copy 120 | **100**\* | 99.2 | 70.4 | 89.4 | 57.8 | **100** | 20.3 |
| Scat. copy 520 | **100**\* | 80.1 | 43.4 | 74.5 | 5.0 | 99.5 | 14.3 |
| Scat. copy 1020 | **99.9**\* | 47.2 | 26.2 | 33.4 | 5.1 | 99.3 | 12.9 |
| Scat. copy 5020 | **99.9**\* | 16.8 | 12.9 | 30.3 | 12.5 | out of mem | 12.9 |
| Add 100 | **0.00003**\* | 0.00019 | 0.00027 | 0.00095 | 0.356 | 0.00103 | 0.420 |
| Add 500 | **0.00031**\* | 0.00032 | 0.02830 | 0.00565 | 0.516 | 0.00291 | 0.536 |
| Add 1000 | 0.00101\* | **0.00066** | 0.04294 | 0.00744 | 0.685 | 0.00638 | 0.537 |
| Add 5000 | 0.00526\* | **0.00074** | 0.12550 | 0.01206 | 1.000 | out of mem | 0.546 |
| pMNIST 784 | 95.4\* | 89.5 | 79.85 | 96.43 | 95.33 | **97.1** | 84.55 |
| pMNIST 3136 | **94.3**\* | 33.6 | 11.7 | 90.31 | 11.7 | 63.5 | 49.7 |

Table 2: Memory size impact. We compute for MemUP method the dependence of `MSE` metric on LSTM layer's dimension in tasks Add 500 and pMNIST 784.

| Task/Memory size | 128 | 256 | 512 | 1024 |
|---|---|---|---|---|
| Add 500 | 0.00028 | 0.00018 | 0.00024 | 0.00056 |
| pMNIST 784 | 89.4 | 92.5 | 95.4 | 96.1 |

**Scattered copy task.** It is similar to the previous one. The only difference is that locations where one has to make predictions are choosen at random in range $[10, \ldots, T-1]$. In such locations $X_i = 1$ and $Y_i$ equals to some element from the first 10 digits. We split this task into four different sub-tasks by the length where $T \in \{120, 520, 1020, 5020\}$. This task is more complex because it requires locations detection and their ordered number counting. As a metric in the previous two tasks we use negative log-likelihood (`NLL`). Train dataset contains 10k sequences and test consists of 1k sequences.

**Add task.** This is an aligned sequence to sequence regression problem $(X \rightarrow Y)$. Each element in $X$ is two dimensional. The first dimension contains a random uniform value $\in [0, 1]$ and second component is 0 or 1. There are only three ones per sequence that correspond to two summands and their sum. Values of $Y$ elements equal to zero except one element at the sum location. `MSE` metric was used here as a loss function and quality measure. Train dataset contains 10k sequences and test consists of 1k sequences. We split this task into four different sub-tasks by the length where $T \in \{100, 500, 1000, 5000\}$.

**Permuted sequential MNIST task (pMNIST).** The dataset is obtained from the ordinary MNIST that includes 60k train images and 10k test images. Each image is flattened to one dimensional vector. It yields vectors of size 784 and we also add zero-value padding to obtain vectors of size 3136. A random permutation changes order of the vector elements consistently. In training procedure we apply `NLL` loss function and measure quality by classification accuracy in %.

## C    Reinforcement Learning Tasks

**Noisy T-Maze.** Namely, the agent starts at the very beginning of the central corridor next to the hint that it observes in the first step of the episode. The agent can only move forward along the corridor. At each step, the agent's observations $o_t$ are represented by a vector of length 3. The first element $o_t[0]$ contains value of the hint: +1 or -1 at the first step, 0 otherwise. The second element $o_t[1]$ is the indicator of reaching the intersection, which is 1 if the agent has reached the place of turn. The last element $o_t[2]$ does not carry information and is a random noise, it is +1 or -1 with equal probability. The agent receives the reward only at the end of the episode. It is +4 if the correct turn was chosen and -3 otherwise.

The length of the maze is defined as the number of steps between the moment the agent sees the hint and the moment the agent has to make a choice based on the value of the hint. The real maze length in experiments varies within 10 steps from episode to episode in order to decorrelate the observations of agents trained in parallel on multiple instances of environments (see PPO algorithms [2], A3C [3], IMPALA [4]). The naming of the environment indicates the minimum possible maze length, for example, the designation T-Maze-LNR-100 means that the maze length in each episode can be from 100 to 109 steps.

**Vizdoom-Two-Colors.** Despite the fact that T-Maze environment allows simulating very long temporal dependencies, it is too simplistic. To explore if MemUP augmentation can be scaled to much more complex tasks and environments it was tested in the Vizdoom-Two-Colors task introduced by Beeching et al. [5]. In this task, the agent is placed in a room filled with acid (see Figure 3.2) and constantly loses health. Objects of green and red colors are scattered throughout the environment. Items of one color replenish the agent's health and give a +1 reward, while others take away health and give a -1 reward. The correspondence between effects and items' colors is determined randomly at the beginning of each episode. This information is conveyed by a column whose color matches the color of health replenishing items. The episode ends when the agent's health drops to zero or after 1050 time steps. At each step agent receives a small "living reward" equal to +0.02. Accordingly, the goal of the agent is to survive as long as possible in the environment by collecting items of the rewarding color.

To solve this problem, it is necessary to keep the color of the signal column in memory in order to be able to select objects of the correct color, even when the column is out of sight. However, in the course of preliminary experiments, it turned out that a reactive agent without memory is able to learn a strategy in which it will always keep the column in sight, even if the room is filled with walls blocking the view from many angles. Therefore, we created a new version of the environment, where the column disappears after the 45th step (see Figure 3.3) and the number of walls in the room is significantly reduced. Thus, the subtask of memorization became harder in comparison with the original version, and the subtask of navigation in the environment was simplified. It is also worth noting that the agent does not receive information about the current health or the rewards received, since these observations actually provide the same information as the color of the column.

## D    T-Maze Scaling Experiments.

In this section, we test MemUP's scaling ability in the T-maze-LNR environment. For each version of the environment (with lengths 500, 1000, 5000, 10000, 20000) we train PPO+MemUP from scratch in 3 separate runs. Picture 1a shows average episodic return after 500 training steps of PPO in the respective environment. Picture 1b shows the root mean squared error of the MemUP memory and predictor modules after 200 training epochs. The evaluation of the memory is conducted in the same way as in Section 8. The memory module is trained with BPTT rollout $r = 20$ in all runs.

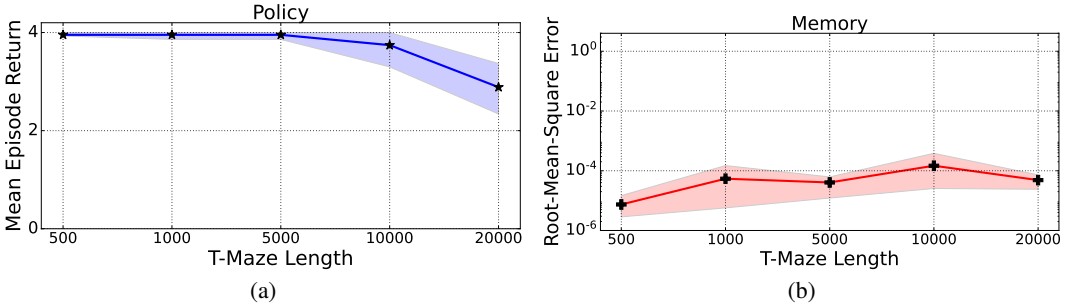

(a)                                          (b)

Figure 1: **T-Maze Scaling Experiments. a)** Final PPO+MemUP performance with respect to corridor length in Noisy T-Maze. Each point is averaged over 3 runs. **b)** Final memory and predictor quality with respect to corridor length in Noisy T-Maze. Each point is averaged over 3 runs.

## E  Open Source Implementation

Implementation for MemUP can be found at: `https://github.com/griver/memup`

## F  Space Complexity for MemUP

Time complexity for MemUP would be the same as for RNNs trained with TBPTT. In our method we focus on space complexity, as MemUP allows RNNs to learn long dependencies with short rollouts.

Space complexity for MemUP is $O(k\frac{r}{f} + r)$, where:

- $r$ - rollout length;
- $f$ - prediction frequency (you can predict from any intermediate step inside a single rollout);
- $k$ - number of prediction targets.

In all experiments (excluding Section 8) frequency (long-term predictions are made at the end of each rollout), therefore the space complexity is only $O(k + r)$. We would like to note that we recommend using $f < r$ in longer rollouts as it saves from the vanishing gradient problem inside a single rollout.

## G  Pseudocode for Memory Training

This section provides a simplified version of the memory training algorithm with feedforward predictor $q_\phi$.

---
**Algorithm 1** Recurrent Memory Training
---
    **Input:** inputs $x$, targets $y$, $r$, $K$, $d_\psi$, $q_\phi$, $g_\theta$
    Train detector $d_\psi$ on the sequence data $x$ and $y$
    Get uncertainty estimates $s$ for each step using $d_\psi$
    $t = 0$ and $m = $ `None`
    **while** $t \leq T$ **do**
       **for** $i = t$ **to** $t + r$ **do**
         $m = g_\theta(x_i, m)$
       **end for**
       $t = \min(T, t + r)$
       Using $s[t : T]$ sample a set $\mathcal{U}_t$ of $K$ elements with highest uncertainty
       Optimize CE loss from eq. 5: $\min_{\theta,\phi} \sum_{k \in \mathcal{U}_t} [-\log q_\phi(y_k|m, x_k)]$
       `stop_gradient(m)`
    **end while**
---

## H  MemUP implementation for RL

### H.1  Uncertainty detector module

The `QRDQN` algorithm[6] is used to train uncertainty detector $d_\psi$. `QRDQN` models the probability distribution over future returns $\mathbb{Z}_\pi(o_t, a_t)$. `QRDQN` represents the distribution with $N$ atoms of the quantile function $\{z_i\}_{i=1}^N$ all having the same probability weights. As proposed by the authors[6] we treat these atoms as samples from the estimated distribution $\mathbb{Z}_\pi(o_t, a_t)$ to estimate the standard deviation:

$$\sigma = \sqrt{\frac{1}{N} \sum_{n=1}^{N} (z_i - \bar{z})^2},$$

where

$$\bar{z} = \frac{1}{N} \sum_{i=1}^{N} z_i.$$

Given standard deviation estimate and assuming that the prediction target is normally distributed, we compute the uncertainty estimation using the formula for the entropy of Gaussian distribution: $H = ln(\sigma\sqrt{2\pi e})$.

Archtecture for the uncertianty detector $d_\psi$:

$$x_t = [E^{obs}(o_t), a_{t-1}]$$
$$z_t = MLP(x_t).$$

Here, $o_t$ is current observation, and $a_{t-1}$ is a previous action. $E^{obs}$ is an observation encoder network. For ViZDoom-Two-Colors $E^{obs} = E^{cnn}$. Convolutional network $E^{cnn}$ consists of 3 convolutional layers: (8, 8) kernel with stride 4 and output channels 32; (4, 4) kernel with stride 2 and output channels 64; (3, 3) kernel with stride 1 and output channels 64. For T-maze environment $E^{obs} = E^{fc}$, where $E^{fc}$ is a single fully-connected (FC) layer with 256 output units. MLP has two linear layers with 512 (256 for T-maze) and $|\mathcal{A}| \times N$ output units respectively.

Table 3: RL: Uncertainty detector training hyperparameters

| HYPER-PARAMETER | T-MAZE-LNR-100 | T-MAZE-LNR-1000 | VIZDOOM-TWO-COLORS |
|---|---|---|---|
| BATCH SIZE | 512 | 512 | 512 |
| LEARNING RATE | 5E-5 | 5E-5 | 5E-5 |
| REPLAY BUFFER SIZE | 1E6 | 1E6 | 1E6 |
| TARGET UPD. INTERVAL | 1E4 | 1E4 | 1E4 |
| NUMBER OF STEPS | 4E6 | 4E6 | 5E6 |
| $N$ | 200 | 200 | 200 |
| $\epsilon_{eval}$ | 0.001 | 0.001 | 0.001 |
| $\epsilon_{train}$ | 1.0 | 1.0 | 1.0 |
| DISCOUNT FACTOR | 0. | 0. | 0.8 |

## H.2  Memory module

Architecture of $g_\theta$ is presented below:

$$e_t = [E^m(o_t), a_{t-1}],$$
$$m_t = LSTM(e_t, m_{t-1}).$$

Here, $E^{obs}$, $E^m$ is an observation encoder ($E^{cnn}$ in Vizdoom-Two-Colors, $E^{fc}$ in T-Maze environments). LSTM module has one layer with hidden size 256.

## H.3  Predictor module

Archtecture for predictor $q_\phi$:

$$e_k = [E^m(o_k), a_k],$$
$$x_k = FC(e_k),$$
$$\hat{y}_k = MLP([m_t, x_k]).$$

Here, index $k \in \mathcal{U}_t$ is a future step selected to train memory using loss from eq. 5. $E^m$ is reused from module $g_\theta$ and shares the same parameters in both modules. FC layer has 256 output units. MLP has two FC layers with 512 and 1 output units respectively. All layers use ReLU nonlinearity.

Training hyper-parameters are shown in Table 4. Hyper-parameters $N$ and $p$ are for sampling training data and the parameter $\gamma$ is for calculating prediction target $R_t$ (see Table 4).

Table 4: RL: Memory and predictor training hyperparameters

| HYPER-PARAMETER | T-MAZE-LNR-100 | T-MAZE-LNR-1000 | VIZDOOM-TWO-COLORS |
|---|---|---|---|
| BATCH SIZE | 128 | 128 | 64 |
| LEARNING RATE | 2E-5 | 2E-5 | 3E-4 |
| NUM. BATCHES PER EPOCH | 300 | 300 | 400 |
| NUM. EPOCHS | 100 | 1000 | 300 |
| ROLLOUT LENGTH | 1 | 1 | 1 |
| $K$ | 1 | 1 | 3 |
| $\gamma$ | 0 | 0 | 0.8 |

## H.4 Agent

We used PPO architecture for the agent. PPO agent recieve a memory state $m_t$ from pretrained module $g_\theta$ as part of its inputs along with current observation $o_t$ and previous action $a_{t-1}$. Weights of $g_\theta$ are frozen during agent training phase. MemUP+PPO had the following archtecture:

$$x_t = E^a_{PPO}(o_t),$$
$$e_t = FC_1([m_t, x_t, a_{t-1}]),$$
$$\pi_t = SoftMax(FC_2(e_t)),$$
$$V_t = FC_3(e_t).$$

The whole neural architecture estimated the policy and value function given the observation with the help of the MemUP memory state. We utilized $E^{cnn}$ for Vizdoom-Two-Colors and $E^{fc}$ for T-Maze tasks as the encoder $E^a_{PPO}$. $FC_1$ net is a FC layer with 256 output units. $FC_2$ is FC layer with $|\mathcal{A}|$ output units. $FC_3$ is a FC layer with 1 output unit.

Training hyper-parameters for MemUP+PPO are listed in Table 5.

Table 5: Hyperparameters for MemUP+PPO agent.

| HYPER-PARAMETER | T-MAZE-LNR-100 | T-MAZE-LNR-1000 | VIZDOOM-TWO-COLORS |
|---|---|---|---|
| BATCH SIZE/NUM. ENVS. | 128 | 128 | 64 |
| LEARNING RATE | 1E-3 | 1E-3 | 1E-3 |
| NUMBER OF STEPS | 5E7 | 6E8 | 1E8 |
| ROLLOUT LENGTH | 5 | 5 | 5 |
| MINI-BATCH SIZE | 4 | 4 | 4 |
| NUM. PPO EPOCHS. | 4 | 4 | 4 |
| $\gamma$ | 0.99 | 0.99 | 0.995 |

# I  MemUP implementation for SL

## I.1  Uncertainty detector module

We estimate uncertainty in SL tasks by values $-\log q_\phi(y_k|\tilde{m}_k, x_k)$. We do not use extra networks here besides memory $g_\theta$ and predictor $q_\phi$. We compute values $\tilde{m}$ with accumulated parameters $\tilde{\theta}$ which we get from the original memory module in every batch iteration:

$$\tilde{\theta} = \alpha\theta + (1 - \alpha)\tilde{\theta},$$

where $\alpha = 0.03$.

## I.2  Memory module

Architecture of $g_\theta$ is presented below:

$$e_t = \text{Embedding}(128)(x_t),$$
$$m_t = LSTM(e_t, m_{t-1}).$$

Here, Embedding is an observation encoder that maps integers into $\mathbb{R}^{128}$. If $x_t$ type is Float then Embedding is two-layer MLP with ReLU activation and size 128. LSTM module has two layers with hidden size 128 and dropout probability 0.1.

## I.3  Predictor module

Predictor is also RNN that learns local context $c_t$ and concatenates it with memory output $m_t$. Architecture of $q_\phi$ is presented below:

$$e_t = \text{Embedding}(128)(x_t),$$
$$c_t = LSTM(e_t, c_{t-1}),$$
$$q_t = MLP(c_t, m_t).$$

Here, Embedding is the same as in memory module. LSTM module has two layers with hidden size 128 and dropout probability 0.1. The final $MLP(c_t, m_t)$ has three linear layers with ReLU activations and one Dropaut layer with probability 0.1. In classification tasks (Copy, pMNIST) we append LogSoftmax to $MLP(c_t, m_t)$.

# J  Resources

We trained our models and baselines using a single server that has:

- 4 CPUs( Intel(R) Xeon(R) CPU E5-2630 v4 @ 2.20GHz) with 10 cores each
- 8 GPUs (GeForce GTX 1080 Ti) with 12GB RAM each
- 256GB RAM