# OpenReview forum: "Explain My Surprise: Learning Efficient Long-Term Memory by predicting uncertain outcomes"
_NeurIPS.cc/2022/Conference — NeurIPS 2022 Accept_

### Official Review · Reviewer_j9Pj · 2022-07-09

**Rating:** 5
**Confidence:** 3
**Soundness:** 3 good
**Presentation:** 4 excellent
**Contribution:** 3 good

**Summary:**

The paper proposes MemUP (Memory for Uncertainty Prediction), a method to learn long-term dependencies for recurrent models. MemUP learns to keep useful past states for future use, so it saves both computation and memory. The memory module is trained to optimize the mutual information between past states and current prediction targets, and only the steps that have highest "uncertainty" would be used for training to further save memory usage. Empirical experiments on supervisied learning and reinforcement learning tasks show that MemUP significantly boost the results over its baselines.

**Questions:**

I have a few questions that are not clarified after reading the paper:

1\. In the uncertainty, how do you get p(y\_t | x\_t), given you are training a model to estimate the p(y\_t | x\_t, m_t)? Is it a separate module (L190)?

2\. For the RL experiments, you set r=1. Can you justify this, given it might hurt the performance of other models? I wonder what the results will be if r is much larger.

3\. In equation 2, the objective function for memory training assumes that the current memory m\_t should be helpful for future prediction at step k, while the m\_t will be updated to m_k at that moment. If my understanding is correct, there is a discrepancy between the objective and inference. Is that right?

**Limitations:**

I do not foresee any potential societal impacts that may occur.

**Strengths And Weaknesses:**

The solution proposed by the paper is valuable for long-sequence modeling. On the one hand, it reduces the computational overhead of RNN model training; on the other hand, it could preserve distant information, which RNNs would easily forget. The methods are simple, and the intuition behind the methods all make sense to me. The paper is well-written, with details clearly formulated and illustrated. Together with its experiments on both supervised learning and reinforcement learning tasks, I can foresee its potential values in the field of reinforcement learning and NLP.

However, the paper does come with some drawbacks and some confusions:

1.  Lack of background work: Though closed related to memory neural networks, none of them is discussed and compared. For example, J Weston, S Chopra, and A Bordes. "Memory networks." ICLR (2014). Similarly, pointer networks should also be discussed, e.g. See et al. (2017). Memory modules are also popular for long-sequence processing in NLP, e.g. coreference resolution (Toshniwal et al. 2020).

2.  Insufficient empirical evidence: Experiments in section 5 are claimed to be long-sequence tasks, while their sequences are not long enough. The lengths of Copy, Scattered copy, and Add are 1000, which can even be handled by some transformers (e.g. T5). The truncation lengths are as small as 10 and up to 60 only, which can be much larger for real applications.

3.  Choice of uncertainty detector: While the description of "Reducing the cost of memory training" in section 2 seems intriguing, the method used in training is simplified into a point estimate of prediction error. In this case, it makes more sense to me to simplify the justification in section 2.

5.  While MemUP uses information theory to help the model to keep the most useful information, the basic idea of MemUP is very similar to LSTM, which purely relies on BPTT to learn how to keep memories. The authors show in sections 5 and 6 that MemUP outperforms LSTM, the baselines are presented in a negative light where the rollout lengths are less than 60 steps. The authors may clearly compare the difference between MemUP and LSTM, as well as providing more evidences to show the advantage of MemUP.

---

> ### Author Response · Authors · 2022-08-02
> **Answer to the Reviewer j9Pj**
>
> Thank you for your feedback! Below, we address your points individually.
>
> > Lack of background work
>
> We argue that Memory Networks and Pointer Networks don't try to solve long-term memory problems. Paper by Toshniwal et al. 2020 does study similar problems but focuses specifically on Document Coreference tasks (we will include it in Related Works).
>
> > Insufficient empirical evidence
> > The lengths of Copy, Scattered copy, and Add are 1000, which can even be handled by some transformers (e.g. T5).
> > The truncation lengths are as small as 10 and up to 60 only
>
> We stopped scaling some tasks upward due to time constraints and our belief that we demonstrated our case. We already have preliminary results demonstrating the ability to train MemUP memory on the T-maze task for a sequence length of 20000 steps and Copy task for sequence length of 5000 steps. We are ready to publish the code that is capable of doing so. We plan to include scaling results for these and other tasks in the updated version of the paper. Short rollouts we will discuss below.
>
> >the method used in training is simplified into a point estimate of prediction error. In this case, it makes more sense to me to simplify the justification in section 2.
>
> In RL experiments we use a different uncertainty detector that directly estimates entropy. Section 2 justifies the use of more sophisticated detectors when simple prediction error is not enough.
>
> >  The authors may clearly compare the difference between MemUP and LSTM, as well as providing more evidences to show the advantage of MemUP.
>
> In the Ablation study (Section B in Supplementary Materials), we compare 3 baselines with  identical architectures and LSTM memory. First Baseline is trained with MemUP, second is trained with  regular TBPTT, and the third is trained by predicting random future events. The results clearly show that purely relying on BPTT to train LSTM is insufficient. In these experiments LSTM trained with MemUP greatly improves the quality of training.
>
>
> > Q1: how do you get $p(y_t | x_t)$, given you are training a model to estimate the $p(y_t | x_t, m_t)$?
>
> Yes. We use a separate module which we call an uncertainty detector (see L114).
>
> > Q2: For the RL experiments, you set $r=1$. Can you justify this, given it might hurt the performance of other models?
> > The authors show in sections 5 and 6 that MemUP outperforms LSTM, the baselines are presented in a negative light where the rollout lengths are less than 60 steps.
>
> In All experiments we have several versions for each baseline. One uses the same rollout length as MemUP (as basically having the same computationally constraints) and the other versions use long rollouts to show their best performance (see pictures 2 and 4). Сonsidering truncated versions of each baseline is essential to demonstrate that using the same budget MemUP allows RNNs to learn long-term dependencies, while baselines decline in performance.
>
> Usually RNNs and Transformers cannot learn temporal dependencies if they never fit inside a single rollout during training ( as can be seen in our experiments). MemUP training allows RNNs to learn temporal dependencies using shorter rollouts. Shorter rollouts save computational space usage ( all intermediate activations inside a single rollout need to be stored). Therefore, we specifically focus on showing that MemUP allows RNNs to learn with shorter rollouts. MemUP can be easily used with longer rollouts (we will include experiments with rollout length of 500 and 1000 in the Supplementary Materials). To fight vanishing gradient problem inside a rollout you can make long term predictions more frequent, e.g. predict from every 10th step (using eq 5.) in rollout of length 1000.
>
> > Q3: If my understanding is correct, there is a discrepancy between the objective and inference. Is that right?
>
> That is not quite right. Memory doesn’t learn to store any hindsight information ($m_t$ is computed from $x_t$ and $m_{t-1}$). Therefore, it can be used in inference.
> Predictor actually receives hindsight information from the future, but also can be used in inference. In MemUP we make a deliberate choice of what to predict (set $U_t$ in eq 3) and when to predict (see Algorithm 1 in Supplementary Materials). This setting can be easily adapted to incorporate current step predictions (with almost no overhead): (1) predict $y_t$ from every step t inside a rollout and (2) from some subset of steps predict Union($\{y_t\}$,  $U_t$). We are training MemUP this way on Copy, Add and Scattered copy tasks. Therefore, the predictor can be used in inference by tasking it to predict $y_t$ from ($m_t$, $x_t$) inputs. We will include this detail in the updated version of the paper.

---

> > ### Comment · Reviewer_j9Pj · 2022-08-08
> > **Response to author response**
> >
> > Thanks for clarifying my concerns and questions.I suggest the authors to include more discussions about the LSTMs and memory networks and publish results on longer sequences and real applications like language modeling.
> >
> >  I take back a some of my points regarding the modeling and raise my score to 5.

---

> > > ### Author Response · Authors · 2022-08-09
> > > **Response #2 to Reviewer j9Pj**
> > >
> > > Thanks for your suggestions and willingness to change mind!
> > >
> > > >I suggest the authors to include more discussions about the LSTMs and memory networks and publish results on longer sequences and real applications like language modeling.
> > >
> > > As a first step we would like to inform you that we have updated the Supplementary Materials by adding code that allows you to train MemUP on longer versions of tasks T-maze (20k steps) and Copy(5k steps) as was claimed in our previous response. This code also contains experiments showing MemUP and policy trained simultaneously for T-maze-1k. We plan to add results and code for all described experiments and their longer versions. There are some technical differences from what was described in the implementation details (Supplementary Materials F. and G.). We will bring them into line, in the updated version of the text.

---

### Official Review · Reviewer_w1z8 · 2022-07-11

**Rating:** 5
**Confidence:** 3
**Soundness:** 2 fair
**Presentation:** 3 good
**Contribution:** 3 good

**Summary:**

In this paper, the authors proposed a method called MemUP to help recurrent neural networks learn long-term dependencies better. Specifically, the authors leverage a memory model to predict future outcomes with high uncertainty. By skipping all states with lower uncertainty, the training process takes less computational cost in backpropagation. The framework involves a uncertainty detector model and a predictor to estimate mutual information used in memory model training. The authors show experimental results on both supervised learning tasks and reinforcement learning tasks with improvement comparing with baselines.

**Questions:**

1. What is the computational cost of the proposed method?
2. By involving auxiliary models to estimate "surprise level", how accurate is the estimation in the proposed framework and experiment?

**Ethics Review Area:**

["I don’t know"]

**Limitations:**

The authors have adequately addressed the limitation and potential negative social impact of their work.

**Strengths And Weaknesses:**

The manuscript has the following strengths:
1. The motivation is reasonable, and the authors proposed a framework to address the motivation.
2. The experiments are solid, the authors provide comparible or better results in comparison with several baselines designated to handle long sequences.
3. For such a complicated framework, the authors are able to illustrate clearly.

The manuscript has the following weaknesses:
1. The proposed framework involves training of several models, including a memory model $g_\theta$, a predictor function $q_\phi$ and a uncertainty detector model $d_\psi$. The whole model is too complicated in my point of view.

Some typos:
1. Title for Figure 2: "final performanse" should be "final performance"

---

> ### Author Response · Authors · 2022-08-02
> **Answer to Reviewer w1z8**
>
> Thank you for your constructive feedback!
>
> We address your questions and comments below.
>
> >The proposed framework involves training of several models, including a memory model , a predictor function  and a uncertainty detector model . The whole model is too complicated in my point of view.
>
> Memory and predictor models essentially behave like a single network during training time (end-to-end training by simply propagating gradients from predictor to memory). Our training process essentially can be described as "run one model and then run the second one focusing on the predictions where the first one failed". It is possible to use memory+predictor network from the previous epoch as a detector (it achieves the same results in most experiments). However, using a separate detector allows for more flexibility and generality. For example, in rl experiments we use QR-DQN for the detector, while predictor+memory simply trains with MSE. We also think that it is easier to train two models performing tasks (uncertainty estimation and correct predictions) than a single model that is good at both.
>
>
> Questions:
> >What is the computational cost of the proposed method?
>
> Time complexity for MemUP would be the same as for RNNs trained with TBPTT.
> In our method we focus on space complexity, as MemUP allows RNNs to learn long dependencies with short rollouts.
>
> Space complexity for MemUP is $O(rk/f + r)$,
> where:
>    - $r$ - rollout length;
>    - $f$ - prediction frequency (you can predict from any intermediate step inside a single rollout);
>    - $k$ - number of prediction targets.
>
> In all experiments (excluding Ablation Study in Supplementary Materials) frequency $f = r$ (long-term predictions are made at the end of each rollout), therefore the space complexity is only $O(k+r)$. We would like to note that
> We recommend using $f < r$ in longer rollouts as it saves from the vanishing gradient problem inside a single rollout.
>
> > By involving auxiliary models to estimate "surprise level", how accurate is the estimation in the proposed framework and experiment?
>
> In T-maze experiments the QR-DQN detector has 100% accuracy (probability of selecting the timesteps connected with long-term dependency).
>
> It is much harder to measure accuracy for vizdoom-two-colors (the agent should be close to an object it is going to pick up), but 50% of the time the detector selects a step that occurred during the last 5 steps before picking up an object.
>
> In SL tasks the detector selects correct timesteps with with a probability of at least 80%.

---

### Official Review · Reviewer_Mqk3 · 2022-07-11

**Rating:** 5
**Confidence:** 3
**Soundness:** 3 good
**Presentation:** 2 fair
**Contribution:** 3 good

**Summary:**

The work introduces memUP, a training method that allows to learn long-term dependencies without backpropagating gradients through the entire sequence (unlike RNNs). The method is based on maximizing mutual information between memory states. Then, it is explained how the equations need and can be approximated. This introduces further assumptions leading to possible vulnerabilities to the Noisy-TV problem. This is further analyzed in the experimental sections. This is further simplified to a practical optimization function related to cross-entropy loss that depends on the memory $m$ at a timestep and the model $q$ given the respective input.
The work then continues by describing how memUP is applied to RNNs. The experimental sections cover results for basic sequence tasks (copy, scattered copy, permuted MNIST, and add task), while comparing to LSTMs, SRNNs, and Transformers. Also, it is experimented in RL settings (T-maze, Vizdoom-Two-colors), showing competitive results to existing methods.



**Questions:**

* How is the optimization of the CE loss (eq 5) is solved for both $\theta$ and $\phi$? When is the stop_gradient(m) applied?
* How would the method needs to be modified to solve regression problems? (for $y_t$ scalar).
* How harder it is to train (and converge) this memory model vs LSTMs with chrono initialization?
* How does this method perform on more real sequential tasks such as language modeling?
* How does the training effort changes when the tasks are more challenging and more parameters could benefit the solution? Is memUp method still viable? If not, what would need to happen?
* What is the effect of the memory size on the solution?
* Would it be possible to measure the effect of the memory on the quality of the solution? For example, what is the performance of the model trained with memUp but with the solution measured with $m_t$ always set to 0 (zero) to "remove" the memory information.


**Limitations:**

The authors describe and analyze the Noisy-TV problem based on their approximation.
They also assume that dependencies are scarce with the increase of the distance.



**Strengths And Weaknesses:**

The work seems original to be the best of this reviewer knowledge and relevant to the NeurIPS community. Enabling models to learn and process long-range dependencies is indeed a problem of interest. In particular, how to train memory augmented models. The works proposes an interesting method to train such models.

The idea behind the method is clear and interesting. The paper focuses on deriving the method, but could have make a better job exploring it further. Instead, authors decided to compare against other methods in two settings (supervised learning on sequences and RL). See questions below. In addition, several of these experiments could have compared to newer recurrent methods, and memory augmented methods (both transformers and RNNs). See [1] and [2] as examples.

The text is clear. Putting more emphasis on how the algorithm is applied to RNNs (and other networks) could really help the reader understand the method and give it more clarity. Defining the tasks in the text instead of the supplementary would be appreciated as well.


[1] Legendre Memory Units: Continuous-Time Representation in Recurrent Neural Networks, Voelker et al., 2019

[2] Not All Memories are Created Equal: Learning to Forget by Expiring, Sukhbaatar et al., 2021

---

> ### Author Response · Authors · 2022-08-02
> **Answer to Reviewer Mqk3**
>
> Thanks for your thoughtful feedback!
> In the following we address it point by point.
>
> > In addition, several of these experiments could have compared to newer recurrent methods, and memory augmented methods.
> > How harder it is to train (and converge) this memory model vs LSTMs with chrono initialization?
>
> Thank you for your suggestions, we will consider running experiments and adding these models to our baseline pool.
> We already have some results (as it was easy to implement) for chrono initialization and full BPTT rollouts:
>
> | Task (Full BPTT) | Accuracy |
> | ----------- | ----------- |
> | copy 120 |   99.7 |
> | copy 520 |   98.4 |
> | copy 1020 |  92.8 |
> | scattered copy 120 |   99.8 |
> | scattered copy 520  | 99.6 |
> | scattered copy 1020 | 98.0 |
> | pMNIST 3136            |  74.1 |
>
> These results show that LSTM(chrono) performs noticeably better than LSTM, but worse than MemUP+LSTM. We would like to note that MemUP can be combined with any RNN architecture (or initalization method) and improve upon it strength.
>
> > How is the optimization of the CE loss (eq 5) is solved for both $\theta$  and $\phi$? When is the stop_gradient(m) applied?
>
> Predictor $q_\phi$ gets two inputs: $x_k$ and $m_t$ (a hidden state produced by memory module $g_\theta$). Therefore, we just propagate gradients from CE Loss (eq. 5) to both predictor and memory modules as if they were a single neural network. We apply stop gradients at the end of each Truncated BPTT rollout. Length of TBPTT is specified in the experiment descriptions (between 1 and 60 in different tasks).
>
> >How would the method needs to be modified to solve regression problems? (for $y_t$ scalar).
>
>  Add task and both RL tasks (memory is trained to predict return R_t which is a scalar) are regression problems. For regression problems we train predictor and memory with MSE loss (L196) which is equal to Cross-Entropy under assumption that $y_t$ has Gaussian Distribution with $\sigma=1$.
>
> >How does this method perform on more real sequential tasks such as language modeling?
> >How does the training effort changes when the tasks are more challenging and more parameters could benefit the solution? Is memUp method still viable? If not, what would need to happen?
>
> We believe that short-term dependencies between tokens greatly outweigh the influence of long-term dependencies in the Language Modeling problems. In that setting Transformers is a king.  Therefore to perform well on these tasks MemUP should be adapted for training Transformers. We have ideas on how to achieve this, but it will require a significantly different training procedure. Therefore, we leave LM problems for future research. The purpose of this work was to demonstrate the usefulness of the main idea for learning long-term depedencies with a method that has sublinear space complexity.  Simple and lightweight RNNs are more suitable for this task.
>
> >What is the effect of the memory size on the solution?
> Increasing memory size improves performance in some tasks:
>
> | Memory size  |    128   |  256   |  512  | 1024 |
> | ----------- | ----------- |----------- |----------- |----------- |
> | Add 500                |   0.00028   |  0.00018   |  0.00024  |   0.00056 |
> | pMNIST 784        |  89.4  | 92.5 |  95.4 |  96.1 |
>
> >Would it be possible to measure the effect of the memory on the quality of the solution? For example, what is the performance of the model trained with memUp but with the solution measured with  always set to 0 (zero) to "remove" the memory information.
>
> Evaluation of pretrained model with $m_t$ set to zero:
>
>  | Task Length  |  Copy      | 	Scattered Copy  |   	Add  |  T-maze |  Vizdoom-two-colors |
> |-------------------|--------------|---------------------------|----------|--------------|----------------------------|
> |  ~1000   |         12.3     |        99.1      |  	0.01006   |    0.5  |    4.1 |
>  |  ~500   |    11.6  |       98.4   | 	0.00430  |     –     |        — |
>  |  ~100  |     12.4      |   92.0    | 0.00065   |  –     |       —  |
>
> In this case the model fails on Copy, Vizdoom-Two-Colors and T-maze tasks and retains decent (but slightly worse) performance in Scattered Copy and Add tasks. While a little surprising these results are reasonable as Scattered Copy and Add contain temporal dependencies that have no lower limit on their length, therefore long-term memory is not as usefull in these tasks.
>
> All results we presented here we will include in the paper or in sapplementary materials.

---

> > ### Comment · Reviewer_Mqk3 · 2022-08-05
> > **Re: Response**
> >
> > Thank you for your response to my questions and executing additional experiments.
> >
> > The experiments are still very limited. Still no comparison to more recent work both in recurrent methods and transformers that tackle long-range dependencies. Most of these papers do compare on real settings beyond simple tasks. This implies bigger datasets as well. For example, Merity et al. [1] can be used to test language modeling at a word level with memUP. Also, there are works that are comparable in performance to this work. For example, [2] obtains similar performance for pMNIST 784 by improving the gating mechanism in LSTMs.
> >
> > Would you please clarify why you thin that "... to perform well on these tasks MemUP should be adapted for training Transformers."?
> > What is the adaptation needed?
> >
> > [1] Regularizing and Optimizing LSTM Language Models, Merity et al., 2017
> > [2] Improving the Gating Mechanism of Recurrent Neural Networks, Gu et al., 2021

---

> > > ### Author Response · Authors · 2022-08-09
> > > **Response #2 to Reviewer Mqk3**
> > >
> > > Thank you for your feedback and interest in our work! We address your responce below.
> > >
> > > > For example, Merity et al. [1] can be used to test language modeling at a word level with memUP.
> > >
> > > Thinking about it, we generally agree that applying MemUP to some RNN architecture suitable for language modeling and demonstrating comparative quality can be a useful experiment. We will conduct additional experiments for language modeling tasks.
> > >
> > > > The experiments are still very limited.
> > >
> > > On the other hand, we do not agree with the statement that our experiments are “very limited”. For example, the paper you cited [1] conducted experiments on: Copy, Add, Sequential Image Classification (pminst  [length=784], CIFAR-10 [len=1024] ), RL problems in the 3D world, and a language modeling task (WikiText103) . In our experiments, we test MemU on tasks: Copy,  Add, Scattered Copy, Sequential Image Classification (pminst  [length=784] and [length=3136]), RL tasks in the 3D world and in the grid world. We have almost identical experimental setups, with the exception of a single language modeling task. But on the plus side, we have a longer version of the experiments in general (also see last paragraph).
> > >
> > >
> > > >Would you please clarify why you thin that "... to perform well on these tasks MemUP should be adapted for training Transformers."? What is the adaptation needed?
> > >
> > > MemUP is a method to decide what to remember by maximizing mutual information between memory's content and future outcomes with high uncertainty.
> > > Applying this idea to RNN is simple: use memory to predict these outcomes (comes from relation between CE and MI) and gradients will modify RNN to store what is important.
> > > This is possible because memory content (hidden state) is a product of RNN network, so we can learn memory by propagating gradients through its content.
> > > In the case of Transformers,  memory content is just a bunch of tokens. Transformer layers are a method of accessing this memory. For most Transformers memory content is decided by using the last N tokens in a sequence. As these Transformers don't decide what to keep and what to remove from their memory, we can't train it the same way as RNN. We have several ideas how to do it (from removing low attention tokens, to estimating MI for each token, etc), but we believe that studying two new training implementations for completely different memory architectures is too voluminous task for one paper.
> > >
> > > We would like to inform you that we have updated the Supplementary Materials by adding code that allows you to train MemUP on longer versions of tasks T-maze (20k steps) and Copy(5k steps) as was claimed in the first response to reviewer j9Pj. This code also contains experiments showing MemUP and policy trained simultaneously for T-maze-1k.  There are some technical differences from what was described in the implementation details (Supplementary Materials F. and G.). We will bring them into line, in the updated version of the text.
> > >
> > > [1] Gu et al. Improving the Gating Mechanism of Recurrent Neural Networks, ICML 2020

---

> > > > ### Comment · Reviewer_Mqk3 · 2022-08-09
> > > > **Thank you for your reply**
> > > >
> > > > I have read your latest response. I'm keeping my current score, based on the lack of language modeling task and missing comparison with more advanced RNNs that can handle this type of tasks.
> > > >
> > > > In the abstract you mention: "This method can be potentially applied to any gradient based sequence learning". You may want to consider updating general claims based on the discussion on transformers above.

---

### Meta-Review · Area_Chair_W1td · 2022-08-27

**Recommendation:** Accept
**Confidence:** Less certain

**Metareview:**

The reviewers found the ideas presented in the paper interesting -- the use of mutual information to train memory for a model, and the clear presentation. Some questions were raised about demonstrating on a more elaborate set up such as NLP tasks -- the main experiments aside from the toy experiments of copy, etc algorithmic tasks, seem to be on RL experiments, but the method has been advertised more broadly in the motivation. Another reviewer raised the question of the complexity of training multiple networks. Nevertheless, the reviewers found the paper interesting enough to recommend a weak accept and I support that recommendation.

From a reviewers lens, I was a little surprised that the paper made no mention of prior works on maximizing mutual information between features of neural networks to improve results. As an example, see the following paper [1] that uses a mutual information regularizer between states at different steps of a recurrent neural networks. There is also a rich literature of doing so for convolutional neural networks. It would have made sense to compare how the idea in the paper performed in comparison to these methods (and in a sense the ablation study which looked at randomly choosing time steps, k, (regardless of the uncertainty estimator) is an experiment in this direction). I understand that part of the paper deals with the choice of time points to increase mutual information between, and so its probably more efficient than the other alternatives, but a comparison (or discussion in related works) would have made the paper stronger.

[1] Better Long-Range DependencyBy Bootstrapping A Mutual Information Regularizer. https://arxiv.org/pdf/1905.11978v1.pdf

**Award:**

No

---

### Decision · Program_Chairs · 2022-09-14

Accept